# Growth and Yield Dynamics in Three Japanese Soybean Cultivars with Plant Growth-Promoting *Pseudomonas* spp. and *Bradyrhizobium ottawaense* Co-Inoculation

**DOI:** 10.3390/microorganisms12071478

**Published:** 2024-07-19

**Authors:** Khin Thuzar Win, Fukuyo Tanaka, Kiwamu Minamisawa, Haruko Imaizumi-Anraku

**Affiliations:** 1Institute of Agrobiological Sciences, National Agriculture and Food Research Organization (NARO), Tsukuba 305-8604, Ibaraki, Japan; 2Research Center for Advanced Analysis, National Agriculture and Food Research Organization (NARO), Tsukuba 305-8604, Ibaraki, Japan; fukuyot@affrc.go.jp; 3Graduate School of Life Sciences, Tohoku University, Sendai 980-8577, Miyagi, Japan; kiwamu.minamisawa.e6@tohoku.ac.jp

**Keywords:** *Bradyrhizobium*, *Pseudomonas* spp., *Glycine max* (L.) Merr., cultivar, inoculation, nodulation, grain yield

## Abstract

Co-inoculation of soybeans with *Bradyrhizobium* and plant growth-promoting bacteria has displayed promise for enhancing plant growth, but concrete evidence of its impact on soybean yields is limited. Therefore, this study assessed the comparative efficacy of two 1-aminocyclopropane-1-carboxylate deaminase-producing *Pseudomonas* species (OFT2 and OFT5) co-inoculated with *Bradyrhizobium ottawaense* (SG09) on the growth, physiology, nodulation efficiency, and grain yield of three major Japanese soybean cultivars: Enrei, Fukuyutaka, and Satonohohoemi. The experiments were conducted in a warehouse under natural light conditions. The treatments included the inoculation of SG09, SG09 + OFT2, and SG09 + OFT5. Compared with *Bradyrhizobium* inoculation alone, co-inoculation led to significant improvements in nodulation efficiency, growth, and physiological performance in the Enrei and Fukuyutaka cultivars, but not in the Satonohohoemi cultivar. Furthermore, co-inoculation significantly boosted the total nitrogen content and ion uptake in the shoots, ultimately leading to a remarkable improvement in the grain yield in the Enrei and Fukuyutaka cultivars. These findings contribute to clarifying the interplay among *Bradyrhizobium*, *Pseudomonas*, and the plant host cultivar. Notably, *Bradyrhizobium*–*Pseudomonas* co-inoculation represents a potentially effective biofertilization strategy for soybean production, highlighting promising avenues for sustainable agricultural practices.

## 1. Introduction

The soybean (*Glycine max* (L.) Merrill) is an economically important grain legume that can establish a symbiotic N_2_-fixing association with soil bacteria belonging to the genera *Bradyrhizobium* and *Sinorhizobium* [1,2]. In legume–*rhizobium* symbiosis, root nodules, in which bacteria convert atmospheric nitrogen into ammonia, provide a sustainable source of plant-available nitrogen and reduce the need for chemical nitrogen fertilizers in agriculture, leading to promote sustainable crop production.

To further enhance soybean–*Bradyrhizobium* symbiosis and boost soybean productivity, creative strategies including co-inoculating *Bradyrhizobium* with other genera of plant growth-promoting bacteria (PGPB), such as *Pseudomonas* [3], *Azospirillum* [4], *Bacillus* [5], and *Serratia* [6], are needed. The beneficial effects of PGPB are mediated by various mechanisms, including the production of diverse metabolites and enzymes by microbes. PGPB also indirectly protect plants against various biotic and abiotic stresses [7,8,9]. The direct mechanisms by which PGPB act on plants include N_2_ fixation, phosphate solubilization, micronutrient solubilization, and the production of one or more phytohormones [10].

In legume–*rhizobium* symbiosis, 1-aminocyclopropane-1-carboxylate (ACC) deaminase (ACCD) and indole acetic acid (IAA) have been frequently suggested as important PGPB components that both promote plant growth [11,12,13] and enhance legume–*rhizobium* symbiosis capacity. ACCD can decrease ethylene levels in higher plants by hydrolyzing ACC, the immediate precursor for ethylene synthesis, into NH_3_ and α-ketobutyrate [14]. In leguminous plants, ethylene is known for its negative effects on nodulation processes [15,16,17,18]. IAA enhances the formation of nodules by altering the physiology and expression of genes associated with nodule initiation in plant cells [19]. IAA also influences rhizobial gene expression by regulating a set of genes involved in the initial stages of symbiosis [20].

Many studies have illustrated that simultaneous infection with specific PGPB increases nodulation and growth in a wide variety of legumes, including soybeans [21,22,23]. Likewise, in our previous study, the *Pseudomonas* species OFT2 and OFT5, which produce both IAA and ACCD, greatly enhanced plant growth and N_2_ fixation in the soybean cultivar Enrei. OFT2 and OFT5 enhanced N_2_ fixation by 76% and 81%, respectively, when co-inoculated with *B. ottawaense* SG09 [24]. However, the PGPB that increase the efficiency of certain rhizobial species are not necessarily identical across soybean cultivars. Consequently, selecting a combination of rhizobia and PGPB with high inoculum efficacy for a particular soybean variety is important to maximize productivity.

In addition, multiple studies have highlighted the positive influence of PGPB on root nodule symbiosis, although concrete evidence of a substantial impact on soybean yields remains elusive [25]. Consequently, it is imperative to confirm the effect of PGPB on soybean yield throughout the entire growth cycle to facilitate the practical application of PGPB in agriculture. To improve N_2_ fixation and grain yield, it is necessary to evaluate various phenotypes of co-inoculation in different soybean cultivars to identify the optimal combination of cultivar, *rhizobium*, and PGPB. Therefore, the present study evaluated the effects of co-inoculating SG09 with the ACCD- and IAA-producing *Pseudomonas* strains OFT2 and OFT5 on plant growth, physiological traits, nodulation, N_2_ fixation, grain yield, and nutrient uptake in three major Japanese soybean cultivars.

## 2. Materials and Methods

### 2.1. Source of Planting Material

The seeds of three major soybean cultivars (Enrei, Fukuyutaka, and Satonohohoemi) were collected from the Genebank Project NARO (https://www.gene.affrc.go.jp/databases-core_collections_jg.php accessed on 12 March 2023). These cultivars are commonly grown in central, southwestern, and northeastern Japan. Before sowing, soybean seeds were sterilized through a 24 h fumigation process using sodium hypochlorite and concentrated hydrochloric acid solutions.

### 2.2. Rhizobial Strains and Growth Conditions

To prepare the inoculum, *B. ottawaense* SG09 was grown for 5 days at 28 °C and 120 rpm in HM (HEPES-MES) broth medium [26] supplemented with 0.1% l-arabinose (*w*/*v*) and 0.025% (*w*/*v*) yeast extract. *B. ottawaense* SG09 was previously isolated from sorghum (*Sorghum bicolor*) grown in Fukushima, Japan. SG09 expresses *nif* and *nod* genes for N_2_ fixation and nodulation and *nos* genes for the reduction of N_2_O to N_2_ [27,28].

### 2.3. PGPB Strains and Growth Conditions

Two PGPB, namely, *Pseudomonas* strains OFT2 and OFT5, were chosen as described by Win et al. [7,9,24]. They were grown in tryptic soy broth at 28 °C and 150 rpm for 24 h.

### 2.4. Greenhouse Experiment

Following seed sterilization, four seeds were planted in Wagner pots (1/5000 acre size) filled with 3 kg of commercial nutrient-free granular soil (Kanuma Sangyo Co., Kanuma city, Japan). Before sowing, the middle layer of the pot was fertilized with 12.5 kg/10 a N, 57.0 kg/10 a P, and 20.0 kg/10 a K in the forms of whole nitrogen (urea formaldehyde, C_2_H_6_N_2_O_2_), monomagnesium phosphate (MgH_2_PO_4_)_2_, and potassium sulfate (K_2_SO_4_), respectively.

The pots were then inoculated with 5 × 10^6^ cells mL^−1^ SG09, which were thoroughly mixed with 500 mL of tap water. The plants were thinned to one uniform plant per pot 4–5 days after sowing, and 1 mL of a PGPB suspension (OFT2 or OFT5) containing approximately 1 × 10^7^ cells mL^−1^ was applied concentrically around the roots of each pot. These pots were cultivated in a warehouse located at the premises of the National Agriculture and Food Research Organization (NARO, Tsukuba, Japan) under natural light conditions. The experimental period was from June 2023 to November 2023. The experimental setup in a randomized block design involved nine replicates for each treatment, which included three soybean cultivars and inoculation with SG09, SG09 + OFT2, and SG09 + OFT5 for two sampling periods. Pots were automatically irrigated nearly at field capacity. Four replicates of the samples were harvested 7 weeks after sowing (7 WAS), with the recorded values including total leaf area (LA), shoot and root dry weight, nodule number, nodule dry weight, and N_2_-fixation activity. The remaining five replicates were harvested to assess seed yield and aboveground biomass.

### 2.5. Measurements of Growth Parameters and Seed Yield

At 7 WAS, the aboveground portions of the plants were divided into stems and leaves. Each leaf was photographed, and LA was calculated using ImageJ (National Institutes of Health, Bethesda, MD, USA). Subsequently, the leaves, shoots, and roots were dried for 72 h in an oven set at 70 °C, and their respective dry weights were then determined. At physiological maturity, plants were harvested, and seed and aboveground biomass were determined.

### 2.6. Chlorophyll Content and Net Photosynthesis Rate

At 7 WAS, before measuring N_2_ fixation, we assessed the relative chlorophyll content (soil plant analysis development (SPAD) value) of the uppermost fully expanded leaves using a soil plant analysis development analyzer (Minolta, Tokyo, Japan). This device measures the transmission of wavelengths absorbed by chlorophylls in intact leaves (mid-position). This determination was performed 12 times for all replicates, and the mean values were calculated for subsequent statistical analyses.

The light-saturated photosynthetic rate (A_sat_) was determined in the second fully expanded leaves from the top of the plant using a high-throughput portable closed-chamber infrared gas analyzer (MIC-100, MASA International Corporation, Tokyo, Japan). The light intensity inside the cuvette was 1200 μmol m^−2^ s^−1^, which is the maximum capacity of this system. The chamber temperature, recorded automatically at the end of each experiment, remained lower than 35 °C. Ambient CO_2_ and humidity conditions were used for the experiment, setting an initial CO_2_ concentration of 370–390 ppm. Measurements were performed 6 WAS on a sunny day from 9:00 a.m. to 11:30 a.m.

### 2.7. N_2_-Fixation Measurement

At 7 WAS, we assessed N_2_-fixation activity through an acetylene reduction assay (ARA) [29]. In this method, the entire root system, along with the nodules of soybean plants, was enclosed in a sealed 100 mL glass vial, in which 10% (*v*/*v*) of the air was replaced with pure acetylene. The samples were then incubated at 25 °C for 20 min to convert acetylene into ethylene. Subsequently, 1 mL of headspace gas was injected into a gas chromatograph (GC-2014 Shimadzu, Kyoto, Japan) equipped with a flame ionization detector and Porapak-N (Shinwa Chemical Industries Ltd., Kyoto, Japan) to measure ethylene gas. Each treatment was replicated four times, with each replicate representing a single plant (*n* = 4). To quantify N_2_ fixation, we expressed the results as nL C_2_H_4_ produced per plant per hour using a standard curve of pure ethylene for reference.

### 2.8. Mineral Ion Analysis and N Uptake

Mineral ion analysis involved quantifying ion content in the shoots at 7 WAS. To achieve this, 20 mg of finely powdered dried samples were digested in 300 μL of concentrated nitric acid, followed by incubation at 95 °C for 3–4 h. After cooling at room temperature for 1 h, the volume was adjusted to 10 mL with Milli-Q water [7]. The mineral ion analysis of plant extracts was performed by inductively coupled plasma mass spectrometry (7700X, Agilent, Santa Clara, CA, USA). The total nitrogen content in the sample was analyzed using a Sumigraph NCH-22F NC analyzer (SCAS, Osaka, Japan).

### 2.9. Data Analysis

The data were analyzed by two-way analysis of variance (ANOVA) using the Statistical Tool for Agricultural Research (STAR) version 2.0.1 (International Rice Research Institute, Los Baños, Philippines) with treatment (PGPB) and variety as fixed factors and the replicates as a random effect. Then, one-way ANOVA was performed to assess the effect of treatment on the measured parameters in each variety. Significant differences between means were compared using Duncan’s multiple range test (DMRT) at a significance level of *p* < 0.05.

## 3. Results

### 3.1. Growth and Leaf Physiology at 7 WAS

Table 1 presents the impact of PGPB on growth and nodulation efficiency at 7 WAS, as well as seed yield and aboveground biomass at harvest, for three soybean cultivars. The influence of PGPB inoculation on leaf growth and physiology was pronounced across all three cultivars. Significant variations in LA, chlorophyll content, and photosynthesis rates were evident according to both plant genotypes and PGPB inoculation (Table 1).

Substantial increases in LA, chlorophyll content, and the photosynthesis rate were observed with co-inoculation with SG09 + OFT2 or SG09 + OFT5 compared with the effects of SG09 inoculation in Enrei and Fukuyutaka (Figure 1 and Figure 2). However, such enhancements were not observed in Satonohohoemi. The physiological aspects of leaves were significantly improved by co-inoculation with SG09 + OFT2 or SG09 + OFT5 compared with SG09 alone in Enrei and Fukuyutaka. The highest SPAD values, as achieved with SG09 + OFT2 or SG09 + OFT5 treatment, resulted in an improved photosynthesis rate in Enrei and Fukuyutaka (Figure 2). Conversely, Satonohohoemi did not exhibit a similar response to PGPB–SG09 co-inoculation. Enrei displayed a substantial increase in total biomass upon SG09 + OFT5 co-inoculation compared with the effect of SG09 inoculation (Figure 1). Notably, a significant interaction between PGPB and variety was observed for LA, photosynthesis rate, and total biomass (Table 1).

### 3.2. Nodulation and Biological N_2_ Fixation

Nodulation, intriguingly, displayed no consistent pattern with the plant genotype, PGPB, or their interaction. The impact of co-inoculating *Bradyrhizobium* and PGPB on nodulation efficiency and N_2_-fixation activity exhibited notable variations among the three cultivars. PGPB inoculation significantly influenced nodule number, and the interaction effect of variety and PGPB was evident for ARA (Table 1).

In Enrei, there were substantial 24% and 41% increases in nodule number upon treatment with SG09 + OFT2 and SG09 + OFT5, respectively, compared with sole SG09 inoculation. The highest nodule dry weight was observed in Enrei after SG09 + OFT5 treatment, surpassing the effect of SG09 inoculation alone. Fukuyutaka displayed modest 14% and 17% increments in nodule number with co-inoculation of OFT2 and OFT5, respectively, compared with SG09 inoculation alone; however, statistical significance was not reached, and no notable difference was observed in nodule dry weight. Notably, in Satonohohoemi, there were no significant differences in nodule number or nodule dry weight among the treatments (Figure 3).

The co-inoculation of *Bradyrhizobium* and PGPB led to considerable variability in N_2_-fixation ability across the cultivars (Table 1, Figure 3). In Enrei, SG09 + OFT2 and SG09 + OFT5 co-inoculation significantly enhanced ARA per plant by 52% and 46%, respectively, compared with sole SG09 inoculation. For Fukuyutaka, only SG09 + OFT2 improved ARA per plant by 31% relative to SG09. Conversely, SG09 + OFT2 had a detrimental effect on ARA in Satonohohoemi, resulting in a significant decrease compared with SG09 alone (Figure 3).

### 3.3. Nutrient Uptake in Soybean Cultivars

The changes in the uptake of macro- and microelements in the shoots of the three soybean cultivars at 7 WAS after co-inoculation with PGPB and *Bradyrhizobium* are illustrated in Table 2. Shoot nutrient uptake significantly differed among the cultivars, PGPB treatments, and their interaction.

Enrei exhibited significant 26% and 36% increases in total nitrogen uptake following SG09 + OFT2 and SG09 + OFT5 co-inoculation, respectively, compared with sole inoculation with SG09. Additionally, the shoot uptake of phosphorus, calcium, magnesium, sulfur, and iron was enhanced by SG09 + OFT5 in Enrei. Likewise, total nitrogen uptake was enhanced in Fukuyutaka following SG09 + OFT2 co-inoculation compared with the single application of SG09. By contrast, Satonohohoemi displayed no discernible variations in total nitrogen uptake across the plants inoculated with SG09 with or without PGPB.

### 3.4. Yield and Aboveground Biomass

Significant varietal disparities were evident in both seed yield and aboveground biomass. The impact of PGPB inoculation on seed yield reached significance (*p* < 0.016), albeit without an interaction effect between variety and PGPB (Table 1).

Notably, Enrei exhibited a substantial 12% increase in seed yield following treatment with SG09 + OFT2 and a significant 14% increase following treatment with SG09 + OFT5 compared with sole SG09 inoculation (Figure 4). Similarly, Fukuyutaka displayed a 20% jump in seed yield with SG09 + OFT2 inoculation. However, no substantial increase in seed yield was observed in Satonohohoemi following co-inoculation with PGPB and *Bradyrhizobium*. The aboveground biomass of the three soybean cultivars was similar among the treatments.

Our findings highlight that the highest co-inoculation effect on seed yield was observed in Fukuyutaka, followed by Enrei, compared to single inoculation with SG09. However, Satonohohoemi cultivar did not exhibit a significant response to co-inoculation.

## 4. Discussion

Scientists have been consistently exploring the idea of simultaneously inoculating legumes with both rhizobia and PGPB. Studies, such as those conducted by Zuffo et al. [4], Tonelli et al. [5], and Bai [6], indicated that co-inoculating *Bradyrhizobium* with various PGPB genera increases soybean productivity. In agreement with previous studies, our study found that co-inoculation of PGPB improves plant growth and leaf physiology as well as nodule numbers in three different soybean cultivars [24]. We examined the effects of *Bradyrhizobium* and PGPB co-inoculation on soybean growth and nodulation efficiency and analyzed whether these effects vary by plant genotype.

Compared with sole SG09 inoculation, combined inoculation of *Bradyrhizobium* and *Pseudomonas* spp. exerted a superior positive impact on plant growth and leaf physiology in Enrei and Fukuyutaka, but not in Satonohohoemi. Similarly, the differential response of chickpea genotypes to co-inoculation of PGPB with N_2_-fixing *Mesorhizobium ciceri* was also described by Imran et al. [30]. It is reported that this bacterium plays a crucial role in promoting plant growth by producing IAA, gibberellins, amino acids, and other polyamines. This in turn stimulates root development, improving water and nutrient absorption by the plants and creating favorable interaction sites between rhizobia and soybeans, as explained by Schmidt et al. [31] and Yadav et al. [32].

In the context of nodule formation, several studies suggested that higher auxin levels in the host plant are essential for nodule formation [33]. Conversely, mutants of the bacterium *B. elkanii* with reduced IAA synthesis produced fewer nodules on soybean roots than the wild-type strain [34]. In a previous paper, we confirmed that the OFT2 and OFT5 strains exhibit high IAA production under in vitro conditions [35]. Taken together, these findings suggest that IAA production by PGPB might have positively influenced the development of nodules [7]. In addition, the plant hormone ethylene and its precursor ACC naturally accumulate in plant roots. Ethylene hinders nodulation at the early stages by regulating the threshold concentration of the Nod factor needed for nodule initiation [36]. Thus, the negative effects of ethylene on the rhizobial infection process might also extend to the symbiotic N_2_-fixation capacity [37]. Bacterial ACCD can control ethylene levels [14,38] by degrading ACC, a precursor of ethylene. The PGPB strains used in the study were also found to produce ACCD, suggesting that the *rhizobium* symbiosis-promoting effects of PGPB might be attributable to its ability to inhibit ethylene synthesis. Our results indicate that co-inoculation of SG09 and PGPB increased the number and dry weight of nodules in Enrei, whereas neutral and negative effects of co-inoculation on the nodulation process were observed in Fukuyutaka and Satonohohoemi, respectively. These results highlight the differential effects of the co-inoculation of *Bradyrhizobium* and *Pseudomonas* spp. on three soybean genotypes. The effects of either IAA synthesized by PGPB or the inhibition of ethylene synthesis by ACCD, or both, on soybeans might have acted on the nodulation process differently in different soybean varieties. Although the differences in the point of action of PGPB inoculation and the mechanisms controlling this action are unknown at present, these phenomena attributable to differences among soybean varieties are of interest.

N_2_-fixation ability (ARA) improved following co-inoculation in both Enrei and Fukuyutaka. However, there were cultivar differences in nodulation and its efficiency for Satonohohoemi. Specifically, the negative impact of SG09 + OFT2 co-inoculation on ARA, compared with SG09 alone, might be because the presence of both strains could trigger physiological changes in the plant. These changes, such as shifts in root architecture, hormone signaling, or resource distribution, can indirectly affect the efficiency of N_2_ fixation by nodules. Further detailed studies on the biochemical and physiological interactions between the co-inoculated strains and the host plant are needed to pinpoint the exact mechanisms. Combined with the fact that no significant differences in seed yield or nitrogen uptake were observed between SG09 + OFT2 and SG09 inoculation in Satonohohoemi, it is possible that the total N_2_-fixation activity of SG09 + OFT2 and SG09 alone during the entire soybean growing season was not significantly different. This significant PGPB × cultivar interaction suggests the need to match certain cultivars with specific strains to improve N_2_ fixation in soybeans. Likewise, other research indicated that the ability of PGPB strains to enhance plant growth differs depending on the specific plant genotype [30,39,40]. In our study, the increased N_2_-fixation activity and higher total nitrogen content in the shoots of Enrei and Fukuyutaka inoculated with *Bradyrhizobium* and *Pseudomonas* spp. indicate that these co-inoculant strains have a strong N_2_-fixation capacity.

Although the co-inoculation of *rhizobium* and PGPB has been predominantly investigated during the initial stages of symbiotic relationship development, there is currently limited reliable information regarding whether co-inoculation technology enhances key traits crucial for achieving high yields across different soybean cultivars. Our findings indicated that the co-inoculation of both symbionts exerts a positive impact on grain yield in soybean cultivars (*p* < 0.017). Specifically, an increase in grain yield was achieved by SG09 + OFT5 co-inoculation in Enrei and SG09 + OFT2 co-inoculation in Fukuyutaka compared with the effects of SG09 alone. Similarly, Sánchez et al. [41] also reported that the effect of *rhizobium*–PGPB co-inoculation on yield was more pronounced than that of single inoculation. Similarly to our studies, Remans et al. [39] also observed that the effect of *Azospirillum* co-inoculation was dependent on the genotype of the common bean.

However, the lack of growth-promoting effects in the Satonohohoemi cultivar following co-inoculation with *Bradyrhizobium* and OFT2 or OFT5 could highlight the importance of considering genotypic variation, compatibility, physiological differences, nutrient requirements, and environmental factors when evaluating the effectiveness of co-inoculation strategies in soybeans. Further research is warranted to elucidate the underlying mechanisms and optimize co-inoculation practices for different soybean cultivars. The findings suggest that differences in host responsiveness to the co-inoculation of *Bradyrhizobium* and *Pseudomonas* spp. between soybean cultivars could have a significant impact on agronomic production.

## 5. Conclusions

Given the substantial variability in the responses of different cultivars to symbiotic partners, it is important to identify soybean cultivars that exhibit a high degree of compatibility in co-symbiosis involving *Bradyrhizobium* and PGPB. This identification will be crucial for recommending soybean varieties to farmers that can maximize the effectiveness of microbial inoculants. Co-inoculating soybeans with *B. ottawaense* SG09 and the present *Pseudomonas* spp. induces higher nodulation, leading to increased grain yield compared with the effects of the sole SG09 inoculation in Enrei and Fukuyutaka. The positive responses of these cultivars to co-inoculation suggest the need for further testing of these cultivars in various field locations alongside these inoculants. This will facilitate the selection of optimal combinations of *Bradyrhizobium* spp. and PGPB to sustainably enhance soybean production.

## Figures and Tables

**Figure 1 microorganisms-12-01478-f001:**
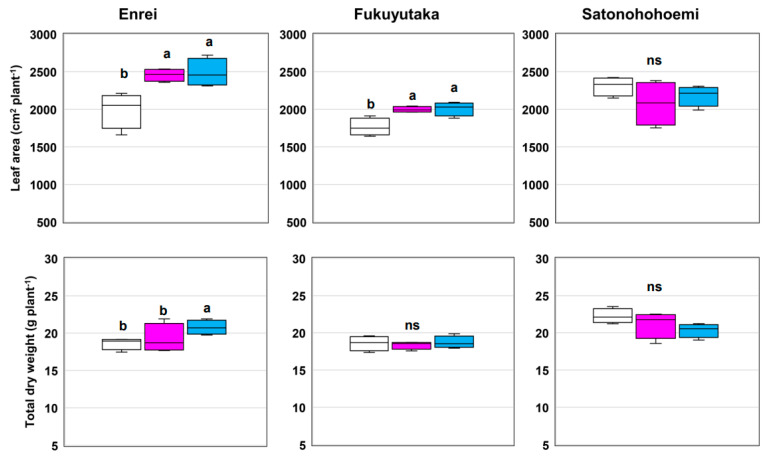
Effects of co-inoculating PGPB with *B. ottawaense* SG09 on leaf area and total dry weight in three different soybean cultivars at 7 weeks after sowing. Different letters indicate classes that exhibit significant differences (*p* < 0.05) using Duncan’s multiple range test. “ns” indicates not significant (*p* > 0.05). The white, pink, and blue boxes represent the treatments SG09, SG09 + OFT2, and SG09 + OFT5, respectively.

**Figure 2 microorganisms-12-01478-f002:**
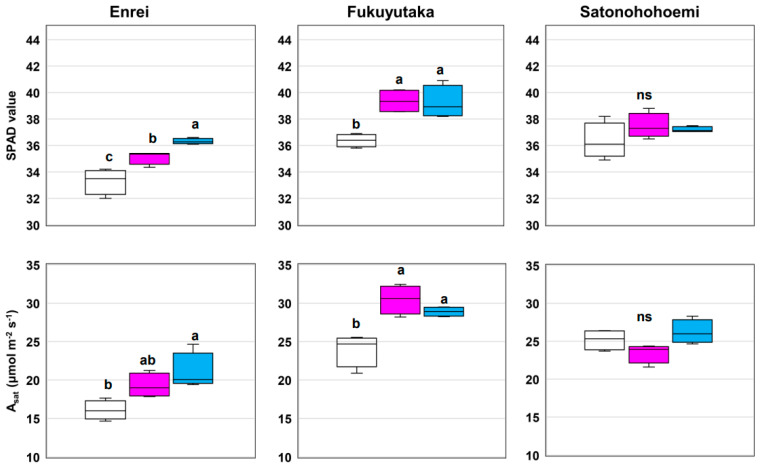
Effects of co-inoculating of plant growth-promoting bacteria with *B. ottawaense* SG09 on the leaf chlorophyll content (SPAD value) and photosynthesis rate (A_sat_) of three different soybean cultivars at 7 weeks after sowing. Different letters indicate classes that exhibit significant differences (*p* < 0.05) using Duncan’s multiple range test. “ns” indicates not significant (*p* > 0.05). The white, pink, and blue boxes represent the treatments SG09, SG09 + OFT2, and SG09 + OFT5, respectively.

**Figure 3 microorganisms-12-01478-f003:**
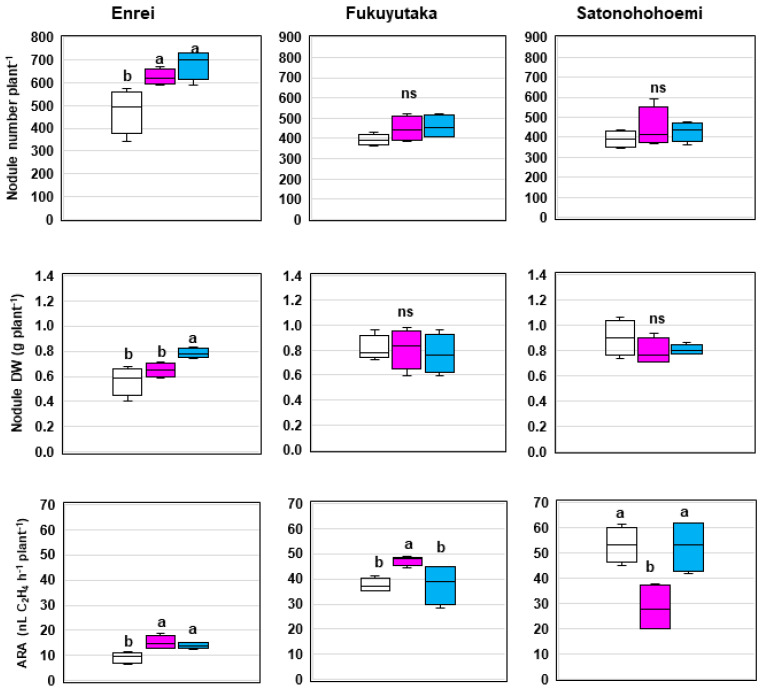
Effects of co-inoculating PGPB and *B. ottawaense* SG09 on nodule numbers, nodule dry weight, and N_2_ fixation (ARA) in three different soybean cultivars at 7 weeks after sowing. Different letters indicate classes that display significant differences (*p* < 0.05) using Duncan’s multiple range test. “ns” indicates not significant (*p* > 0.05). The white, pink, and blue boxes represent the treatments SG09, SG09 + OFT2, and SG09 + OFT5, respectively.

**Figure 4 microorganisms-12-01478-f004:**
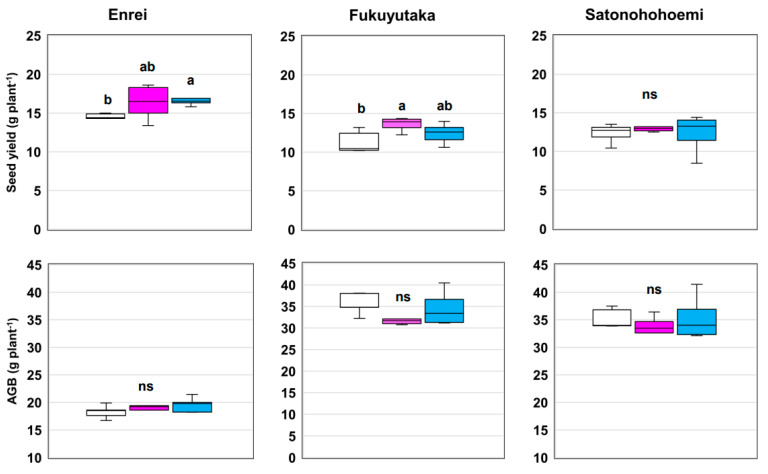
Effects of co-inoculating plant growth-promoting bacteria with *B. ottawaense* SG09 on seed yield and aboveground biomass in three different soybean cultivars at maturity. Different letters indicate classes that display significant differences (*p* < 0.05) using Duncan’s multiple range test. “ns” indicates not significant (*p* > 0.05). The white, pink, and blue boxes represent the treatments SG09, SG09 + OFT2, and SG09 + OFT5, respectively.

**Table 1 microorganisms-12-01478-t001:** Results of two-way ANOVA of growth, nodulation, and physiological parameters at 7 weeks after sowing.

ANOVA (*p*)			
Traits	Variety	PGPB	Variety × PGPB
Leaf area	**0.0004**	**0.0307**	**0.0086**
SPAD values	**0.0000**	**0.0000**	0.1130
Photosynthesis rate (A_sat_)	**0.0000**	**0.0000**	**0.0026**
Total biomass	**0.0032**	0.7314	**0.0312**
Nodule number	**0.0003**	**0.0009**	*0.0985*
Nodule dry weight	**0.0155**	0.7012	*0.0875*
Acetylene reduction assay (ARA)	**0.0002**	*0.0989*	**0.0000**
Seed yield	**0.0003**	**0.0168**	0.4800
Aboveground biomass	**0.0000**	0.1404	0.4840

*p*-values of significant effects of variety and inoculation and the interaction between them are highlighted in bold. Italic values denote significance at a 10% level.

**Table 2 microorganisms-12-01478-t002:** Effects of co-inoculating plant growth-promoting bacteria and *rhizobium* (SG09) on nutrient uptake in the shoots (mg per plant) of Enrei, Fukuyutaka, and Satonohohoemi cultivars at 7 weeks after sowing. Values are presented as means ± standard deviation (n = 4).

Cultivar	Treatment	N	P	K	Ca	Mg	S	Cu	Fe
Enrei	SG09	256.9 ± 28.7	1584.1 ± 361.8	25,692.0 ± 5312.4	17,884.3 ± 3394.9	4793.9 ± 842.8	2051.9 ± 463.4	2.8 ± 0.6	85.7 ± 3.0
	SG09 + OFT2	319.8 ± 14.0 **	1990.1 ± 161.7	29,209.9 ± 2282.4	20,732.5 ± 1802.0	5412.5 ± 389.8	2381.6 ± 176.4	3.2 ± 0.10	85.2 ± 5.5
	SG09 + OFT5	348.1 ± 12.7 **	2156.8 ± 83.4 *	32,155.4 ± 397.8	24,364.6 ± 362.0 *	6018.4 ± 448.2 *	2698.6 ± 95.2 *	3.6 ± 0.4	101.3 ± 7.5 *
Fukuyutaka	SG09	286.1 ± 18.9	2007.5 ± 41.8	25,688.9 ± 2589.2	21,899.5 ± 730.2	4957.7 ± 48.4	2507.8 ± 149.2	2.9 ± 0.1	88.0 ± 19.6
	SG09 + OFT2	334.6 ± 7.0 *	1879.8 ± 203.4	26,233.9 ± 2363.0	19,079.9 ± 1909.8	4293.4 ± 479.6	2434.1 ± 264.1	2.8 ± 0.2	87.8 ± 8.3
	SG09 + OFT5	306.8 ± 16.5	1901.7 ± 104.6	28,183.5 ± 353.0	18,786.0 ± 476.7 **	5000.3 ± 76.6	2506.6 ± 68.6	3.1 ± 0.1 *	91.8 ± 12.1
Satonohohoemi	SG09	290.1 ± 19.2	1981.4 ± 144.4	32,187.1 ± 1547.8	25,138.2 ± 807.7	6431.2 ± 497.9	2898.4 ± 13.2	3.4 ± 0.2	89.3 ± 10.8
	SG09 + OFT2	279.5 ± 37.0	2110.0 ± 103.5	31,852.4 ± 4623.6	23,278.8 ± 3835.5	6090.6 ± 1151.2	2749.1 ± 279.5	3.4 ± 0.6	87.7 ± 17.2
	SG09 + OFT5	280.5 ± 26.9	1892.5 ± 195.7	31,628.1 ± 573.3	24,891.4 ± 652.5	6298.3 ± 194.7	2846.7 ± 285.1	3.5 ± 0.4	108.4 ± 10.9 *

*p* < 0.01 **, *p* < 0.05 *.

## Data Availability

All data generated or analyzed during this study are included in this published article.

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
