# Peer review of "Growth and Yield Dynamics in Three Japanese Soybean Cultivars with Plant Growth-Promoting Pseudomonas spp. and Bradyrhizobium ottawaense Co-Inoculation"

_microorganisms, 2024, doi:10.3390/microorganisms12071478_

Round 1
Reviewer 1 Report
Comments and Suggestions for Authors
The manuscript titled "Growth and yield dynamics in three Japanese soybean cultivars after plant growth-promoting Pseumodonas spp. and Bradyrhizobium ottawaense co-inoculation" contains interesting research results for science and agricultural practice.
I have read the publications with interest.
The text of the manuscript is well written. I included some comments in the original PDF text
General notes:
correct affiliation as required by Microorganisms journal or view recent issues
write the full name of Bradyrhizobium ottawaense in the abstract
in keywords add: Glycine max (L.) Merr., cultivar, inoculation
line 179 replace "leaf color" with "Soil Plant Analysis Development"
explain the abbreviation "ns" under the figures
explain under table 2 what ± means
After corrections, I recommend publishing the manuscript in the journal Microorganisms
I hope that my comments will help the authors improve the text of the manuscript.
Thank you for your cooperation.

Author Response
|
No. |
Comments |
Responses |
|
1 |
The manuscript titled "Growth and yield dynamics in three Japanese soybean cultivars after plant growth-promoting Pseumodonas spp. and Bradyrhizobium ottawaense co-inoculation" contains interesting research results for science and agricultural practice. I have read the publications with interest. The text of the manuscript is well written. I included some comments in the original PDF text |
Thank you very much for your constructive comments. We have carefully followed all suggestions and revised the manuscript accordingly. |
|
2 |
General notes: correct affiliation as required by Microorganisms journal or view recent issues |
We have followed the suggestions and made the revisions accordingly. |
|
3 |
write the full name of Bradyrhizobium ottawaense in the abstract |
Thank you for your comment. We have revised it accordingly in Line 21. |
|
4 |
in keywords add: Glycine max (L.) Merr., cultivar, inoculation |
Thank you for your comment. We have incorporated them accordingly in Line 33. |
|
5 |
line 179 replace "leaf color" with "Soil Plant Analysis Development" |
Thank you for your comments. SPAD first appeared in L130. Accordingly, we have added "Soil Plant Analysis Development" in the relevant line (L:130). |
|
6 |
explain the abbreviation "ns" under the figures |
The explanation for the abbreviation ‘ns’ has been added under the figures. |
|
7 |
explain under table 2 what ± means |
Thank you for your comments. We have added the explanation of "±" under Table 2 accordingly in Line 253. |
Reviewer 2 Report
Comments and Suggestions for Authors
Article title:
“Growth and yield dynamics in three Japanese soybean cultivars after plant growth-promoting Pseumodonas spp. and Bradyrhizobium ottawaense co-inoculation".
Major Comments:
- The manuscript is traditional (there is nothing novelty in it) and the aim of ​​the manuscript is based on another paper published in 2023 with one of the tested soybean varieties.
- Why was the inoculation process performed separately for each bacteria (B. ottawaense and Pseumodonas?
Mainor Comments
Title:
- Please modify it again and remove the word (after).
Abstract:
- Line 22: write the soybean cultivars used.
- Mention the rates of increase in nodule formation and growth in co-inoculation treatments compared to individual inoculation
- Arrangement of the studied cultivars in terms of their increased growth (Growth and yield dynamics) under co-inoculation conditions
Introduction:
- Line 37, 51, 75, 247, 299 and 329: rhizobium (italic).
- The authors highlighted the production of growth regulators for PGPR, such as the production of IAA and ACCD, and this is what was found in other papers and was not conducted in this manuscript. Therefore, these parts must be removed and the focus must be on presenting the role and efficiency of co-inoculation in improving the growth dynamics of different soybean varieties.
Materials and Methods
- Line 89: HM broth medium (Write completely).
- Line 100: The seeds (remove).
- Mention the number of seeds that were planted in the pot before the thinning process
- Mention the amount of inoculation used with Bradyrhizobium bacteria
- Line 106: after seed sowing (remove).
Results
- Table 1: Added the abbreviation below the Table i.e. ARA.
- Table 2: Added the units for the nutrient uptake.
Discussion
- Acceptable
Conclusion
- Delete lines 350-352.
Author Response
|
No. |
Comments |
Responses |
|
1 |
The manuscript is traditional (there is nothing novelty in it) and the aim of ​​the manuscript is based on another paper published in 2023 with one of the tested soybean varieties. |
Thank you very much for the constructive comments. We have revised the manuscript accordingly. In our previous report, we demonstrated that co-inoculation of Bradyrhizobium ottawaense SG09 with two PGPB strains improved the growth and nodulation efficiency of the ‘Enrei’ variety. As noted in lines 67-68, the PGPB that enhance rhizobial efficiency are not necessarily the same across different soybean cultivars. To investigate this further, we studied three major soybean cultivars commonly grown in central, southwestern, and northeastern Japan. Multiple studies have highlighted the positive influence of PGPB on root nodule symbiosis, though concrete evidence of their significant impact on soybean yields remains limited. Co-inoculation has emerged as a promising strategy for enhancing the growth and yield of crops. Specifically, the combined inoculation with Bradyrhizobium and plant growth-promoting bacteria (PGPB) has shown benefits, but its effects on soybean yields have not been fully confirmed. Therefore, this study investigated the effects of co-inoculating B. ottawaense SG09 and Pseudomonas spp. on plant growth, physiological traits, nodulation, N2-fixation, grain yield, and nutrient uptake in three major Japanese soybean cultivars. We believe this study provides valuable insights and a broader understanding of the co-inoculation effects across different soybean cultivars, thereby addressing the gaps highlighted in previous research. |
|
2 |
Why was the inoculation process performed separately for each bacteria (B. ottawaense and Pseumodonas? |
The inoculation process was performed separately for each bacterium (B. ottawense and Pseudomonas) to ensure the optimal concentration and performance of each. This approach also avoids resource competition between the bacteria at the first seedling emergence (4-5 days). Additionally, separate inoculation allows for a better understanding of the synergistic effects when both bacteria are present. |
|
3 |
Please modify it again and remove the word (after). |
Thank you for your comments and suggestions. Accordingly, I have revised the title as follows: Line 2-4: ‘Growth and yield dynamics in three Japanese soybean cultivars with plant growth-promoting Pseudomonas spp. and Bradyrhizobium ottawaense co-inoculation’ |
|
4 |
Line 22: write the soybean cultivars used. |
We have included the names of the soybean cultivars in the relevant line (Line 22) accordingly. |
|
5 |
Mention the rates of increase in nodule formation and growth in co-inoculation treatments compared to individual inoculation |
Due to journal guidelines limiting abstract word count and the numerous treatment combinations tested, we have provided detailed percentage increases in nodule formation and growth rates in the results section. |
|
6 |
Arrangement of the studied cultivars in terms of their increased growth (Growth and yield dynamics) under co-inoculation conditions |
Please refer to response No. 5 and review the results section for details on the arrangement of the studied cultivars in terms of their increased growth under co-inoculation conditions in Lines 267-269. |
|
7 |
Line 37, 51, 75, 247, 299 and 329: rhizobium (italic). |
"Rhizobium" has been italicized throughout the manuscript as per your suggestions. |
|
8 |
The authors highlighted the production of growth regulators for PGPR, such as the production of IAA and ACCD, and this is what was found in other papers and was not conducted in this manuscript. Therefore, these parts must be removed and the focus must be on presenting the role and efficiency of co-inoculation in improving the growth dynamics of different soybean varieties. |
We have deleted it accordingly. |
|
9 |
Line 89: HM broth medium (Write completely). |
We have added it accordingly in Line 92. |
|
10 |
Line 100: The seeds (remove). |
We have removed it accordingly. |
|
11 |
Mention the number of seeds that were planted in the pot before the thinning process |
We have detailed this information in Line 101 of the manuscript. |
|
12 |
Mention the amount of inoculation used with Bradyrhizobium bacteria |
We have detailed this information in Line 107 of the manuscript. |
|
13 |
Line 106: after seed sowing (remove). |
We have removed it accordingly. |
|
14 |
Table 1: Added the abbreviation below the Table i.e. ARA. |
Detailed information about ARA has been added in Table 1. |
|
15 |
Table 2: Added the units for the nutrient uptake. |
Units for the nutrient uptake have been added accordingly. |
|
16 |
Delete lines 350-352. |
The relevant parts have been deleted as per your suggestions. |
Round 2
Reviewer 2 Report
Comments and Suggestions for Authors
Accept in present form